# DON'T WASTE MISTAKES: LEVERAGING NEGATIVE RL-GROUPS VIA CONFIDENCE REWEIGHTING

## ABSTRACT

Reinforcement learning with verifiable rewards (RLVR) has become a standard recipe for improving large language models (LLMs) on reasoning tasks, with Group Relative Policy Optimization (GRPO) widely used in practice. Yet GRPO wastes substantial compute on negative groups: groups in which no sampled response is correct yield zero advantage and thus no gradient. We ask whether negative groups can be leveraged without extra supervision. Starting from a maximum-likelihood (MLE) objective in reward modeling, we show that the MLE gradient is equivalent to a policy gradient for a modified value function. This value function adds a confidence-weighted penalty on incorrect responses, imposing larger penalties on more confident mistakes. We refer to this as **L**ikelihood **E**stimation with **N**egative **S**amples (**LENS**). LENS modifies GRPO to assign non-zero, confidence-dependent rewards to incorrect generations, making negative groups informative and converting previously wasted samples into useful gradient updates. On the MATH benchmark with Llama-3.1-8B and Qwen-2.5-3B, the proposed variant consistently outperforms GRPO baseline, with significant gains on harder items. These results demonstrate a principled and practical way to "rescue" negative groups, improving efficiency and performance in RLVR.

## 1 INTRODUCTION

Large language models (LLMs) fine-tuned with reinforcement learning and verifiable rewards (RLVR) (Shao et al., 2024; Guo et al., 2025) have shown strong gains on complex reasoning tasks, with algorithms such as Group Relative Policy Optimization (GRPO) (Shao et al., 2024; Guo et al., 2025) emerging as practical defaults. A persistent inefficiency, however, is how these methods handle negative groups—the generation group in which no sampled response is correct. In GRPO and its variants, such groups contribute zero advantage and therefore no gradient signal. This is especially common at the start of training and on harder reasoning problems, where negative groups can constitute a substantial fraction of compute, effectively wasting already-generated trajectories.

We therefore ask: can we learn from negative groups without additional supervision in a *principled* way? Our starting point is deliberately simple: to learn from negative groups, the natural approach is reward modeling that distinguishes correct from incorrect answers, optimized with maximum likelihood (MLE). From this likelihood perspective, the MLE gradient is equivalent to a policy gradient on a modified RLVR value function. The modified value adds a confidence-weighted penalty for incorrect responses: the more confident the model is in a wrong answer, the larger the penalty. Intuitively, it discourages overconfident failure modes, thereby encouraging exploration of lower-probability yet plausible alternatives.

This equivalence lets us modify GRPO directly. It yields a drop-in change in which incorrect generations receive non-zero, confidence-dependent rewards (i.e., lower rewards when confidence is higher). As a result, negative groups now provide informative advantage estimates, converting previously wasted samples into useful gradient updates and promoting exploration on hard negatives. We term this algorithm *LENS*: *Likelihood Estimation with Negative Samples*.

We evaluate LENS on mathematical reasoning using the MATH benchmark with `Llama-3.1-8B-Instruct` and `Qwen-2.5-3B-Base`. In both settings, our GRPO variant consistently outperforms the GRPO baseline across all Pass@$k$ metrics. Stratifying by difficulty, we find that gains are concentrated on the Levels 4-5 subsets (hard items), consistent

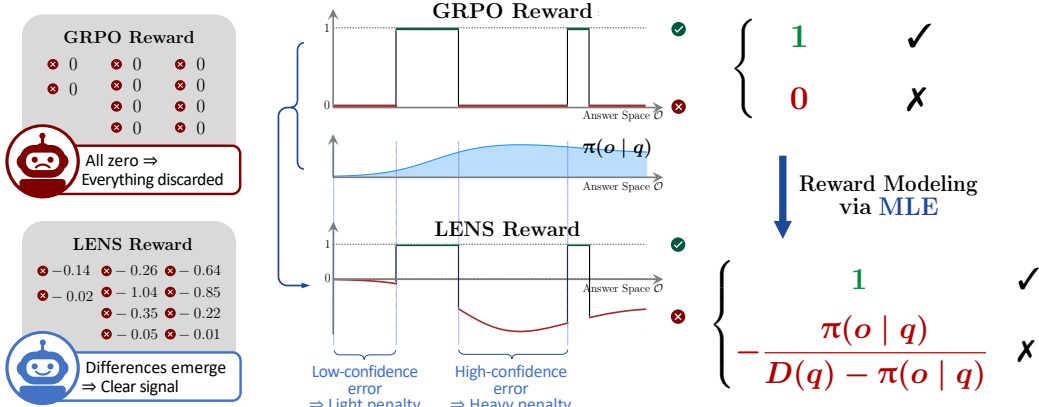

Figure 1: **Overview of our approach.** Standard approaches like GRPO assign a uniform reward of $0$ to all incorrect answers. This provides no learning signal, causing these samples to be discarded. Our method, LENS, is derived from reward modeling via Maximum Likelihood Estimation (MLE) and assigns non-zero, confidence-dependent rewards to incorrect responses. This creates a clear learning signal where differences emerge from the samples, converting previously discarded information into useful gradient updates.

with repurposed negative groups driving increased exploration for hard questions. We train on two distinct math training datasets to demonstrate the generality of our method.

We summarize our contributions as follows:

- We introduce a likelihood framework, *Likelihood Estimation with Negative Samples (LENS)*, that *explicitly connects* reward modeling and policy optimization.

- LENS yields a principled value function whose additional term penalizes *overconfident incorrect* answers, formalizing how negative-group signals should be used and calibrated within the objective.

- We propose a GRPO variant that assigns *non-zero, confidence-dependent* rewards to incorrect generations, thereby leveraging negative groups rather than wasting them. It is plug-and-play with *negligible* computational overhead.

- Empirical results support our algorithm's effectiveness and show increased exploration, as reflected in Pass@$k$.

## 2 RELATED WORK

*RLVR.* Recent work has shown that reinforcement learning (RL) can effectively refine LLMs for reasoning. In RLVR, the LLM is treated as a policy that generates a chain-of-thought (CoT) reasoning process, and it receives a deterministic reward based on whether the final answer can be algorithmically verified. Recent works (Shao et al., 2024; Guo et al., 2025; Team et al., 2025) show that RLVR can elicit emergent reasoning behaviors and dramatically boost math and coding performance compared to the base model. Underlying most of these RLVR methods is the Group Relative Policy Optimization (GRPO) algorithm (Shao et al., 2024). GRPO is an efficient variant of Proximal Policy Optimization (PPO) (Schulman et al., 2017) that drops the value network and instead computes advantages from grouped outputs. In this way, with a group of all incorrect generations, the advantage is $0$, and these groups do not contribute to the optimization. In this work, we try to make use of these negative groups.

*Learning from negatives.* Recent work has emphasized that negative samples are not merely noise but a useful training signal in LLM reasoning. One direction explores asymmetric treatment of positives and negatives in REINFORCE-style training: Roux et al. (2025) introduce an asymmetric variant of importance sampling to speed up learning. Arnal et al. (2025) demonstrate that asymmetric REINFORCE, and in particular reducing the signal from negative samples, can be beneficial when data is off-policy. Lyu et al. (2025) propose to reweight positive and negative samples at the

token level using a learned reward model combined with log-likelihood. Zhu et al. (2025) demonstrate that training only on negatives, assigning reward $-1$ to incorrect and $0$ to correct answers, can outperform baselines on Pass@$k$ for large $k$.

Another line of work argues that entirely wrong completions may still contain valuable sub-signals. Chen et al. (2025a) assign fractional rewards within all-negative groups, Yang et al. (2025) mine correct sub-steps from long chains of thought, and Li et al. (2024b) leverage negative rationales through a dual-LoRA distillation framework. These methods demonstrate that even within incorrect trajectories, certain steps are worth reinforcing, particularly in long reasoning traces where correct and incorrect steps alternate. A key drawback of these approaches is that evaluating intermediate reasoning steps is labor-intensive, and accurate automation remains underexplored.

Our contribution is to provide a framework that stratifies reward signals within negative samples using only outcome rewards and probability, balancing computational efficiency with the benefits of learning from structured negatives.

## 3   PRELIMINARIES AND MOTIVATION

We start with background on policy optimization and the motivation for our method.

### 3.1   LANGUAGE MODEL REASONING AS POLICY OPTIMIZATION

We begin with a basic setting: given a question $q \in \mathcal{Q}$, a language model $\pi$ is tasked with generating an answer $o \in \mathcal{O}$. To evaluate correctness, we assume the existence of a reward function $r^\star : \mathcal{Q} \times \mathcal{O} \to \{0,1\}$, which assigns $1$ if the answer $o$ is correct for the given question $q$, and $0$ otherwise.

The ultimate goal of training the language model is to improve its accuracy rate. Formally, this corresponds to maximizing the expected reward:

$$\text{maximize}_\pi \quad J(\pi) \ := \ \mathbb{E}[r^\star(q, o)], \qquad \text{where } q \sim \xi, o \sim \pi(\cdot \mid q). \tag{1}$$

Here $\xi$ denotes the distribution of questions. Equation (1) is the central criterion: it asks us to design a policy $\pi$ that maximizes the expected correctness of generated responses.

### 3.2   MOTIVATION: NEGATIVE GROUPS IN RLVR

In practice, *Group Relative Policy Optimization (GRPO)* has become a default algorithm for optimizing LLM reasoning ability for the objective in Equation (1). Concretely, for each verifiable question $q$, we draw a group of $G$ candidates $\{o_i\}_{i=1}^{G} \sim \pi_{\theta_{\text{old}}}(\cdot \mid q)$, obtain scalar rewards $r_i := r^\star(q, o_i) \in \{0, 1\}$, and form zero-mean, unit-variance group advantages

$$\widehat{r}_i \ = \ \frac{r_i - mean(\{r_j\}_{j \in [G]})}{std(\{r_j\}_{j \in [G]})}. \tag{2}$$

With outcome-only rewards, the same advantage $\widehat{A}_{i,t} = \widehat{r}_i$ is assigned to all tokens $t$ in response $o_i$. GRPO then maximizes a clipped PPO-style surrogate with an explicit per-token KL regularizer to a fixed reference $\pi_{\text{ref}}$:

$$J_{\text{GRPO}}(\pi_\theta) = \mathbb{E}_{q, \{o_i\}} \frac{1}{G} \sum_{i=1}^{G} \frac{1}{|o_i|} \sum_{t=1}^{|o_i|} \Big[ \min\big(\rho_{i,t}\widehat{A}_{i,t}, \ \text{clip}(\rho_{i,t}, 1 - \epsilon, 1 + \epsilon)\widehat{A}_{i,t}\big) \Big], \tag{3}$$

where $\rho_{i,t} := \frac{\pi_\theta(o_{i,t}|q, o_{i,<t})}{\pi_{\theta_{\text{old}}}(o_{i,t}|q, o_{i,<t})}$ is the correction for off-policy samples. We omit the KL divergence term following the common practice as $\beta = 0$.

GRPO is a practical policy-gradient method for LLMs because it computes advantages from *group-relative* statistics rather than a learned value function (critic). This makes it simple and robust for long-form reasoning, where sequences are long and rewards arrive only after a complete solution.

However, GRPO wastes substantial compute on negative groups. If an entire group is incorrect, i.e., all rewards $\{r_i\}$ are zero, the advantages collapse to zero, yielding no contribution to the policy gradient. Figure 2 shows the fraction of all-negative groups during training with group size $G = 16$: despite 16 generations per prompt, nearly 45% of groups are all-negative early in training, and about 35% remain even by the end. These groups consume substantial generation compute yet contribute no learning signal.

Figure 2: Negative group ratio during GRPO training of `Llama-3.1-8B-Instruct` with MATH and Numina 1.5. $G = 16$.

## 4 A LIKELIHOOD-BASED FRAMEWORK FOR REASONING

We now seek to find a principled framework to use the negative groups. A direct route is reward modeling: train a model to discriminate correct from incorrect responses. We develop a likelihood-based formulation of reward modeling and show how it connects to policy optimization.

### 4.1 FROM POLICY LEARNING TO REWARD MODELING

While our goal is to optimize the policy, the task becomes clearer when re-expressed through reward modeling. To illustrate this connection, we turn to a simple multiple-choice example.

**Illustrative Example: Multiple-Choice Reasoning.** Suppose a single question $q$ comes with six possible answers: $A, B, C, D, E, F$. Out of these, only $A$ and $B$ are correct. We can think of an unknown ground-truth probability function

$$p^\star(q, o) = \mathbb{P}\big[\text{Answer } o \text{ is correct for question } q\big].$$

For math problems, this function is deterministic: each answer is either correct ($p^\star = 1$) or incorrect ($p^\star = 0$) and $p^\star = r^\star$. More generally, however, $p^\star$ could take fractional values in $[0, 1]$ to reflect varying confidence or partial correctness.

In this example, the desirable optimal policy $\pi^\star$ for Equation (1) is one that selects only from the correct options. For instance:

$$\pi^\star(A \mid q) = \pi^\star(B \mid q) = \tfrac{1}{2}, \quad \pi^\star(C \mid q) = \cdots = \pi^\star(F \mid q) = 0.$$

This $\pi^\star$ randomly chooses between the correct answers $A$ and $B$.[1] This relationship can be expressed more generally as

$$p^\star(q, o) = \frac{1}{D(q)} \pi^\star(o \mid q), \tag{4}$$

where $D(q)$ is a normalizing factor defined by

$$D(q) = \left\{ \sum_{o \in \mathcal{O}} p^\star(q, o) \right\}^{-1}. \tag{5}$$

Intuitively, $D(q) \in (0, 1]$ captures the *difficulty* of the question. If only one answer is correct, $D(q) = 1$, indicating a hard question. If multiple answers are correct, $D(q)$ becomes smaller, signaling an easier question.

In practice, we do not have direct access to the full probability function $p^\star$. Instead, we observe data samples of the form $(q, o, r)$, where $r \sim \text{Bernoulli}\big(p^\star(q, o)\big)$. Reward modeling then fits a model $p_\theta$ to these observations to approximate $p^\star$. Through the relation in Equation (4), we can recover one optimal policy $\pi^\star$. Therefore, policy learning reduces to the statistical task of estimating reward probabilities.

---

[1]Here we select an optimal policy that chooses uniformly at random among all correct answers. In more general settings we may have preferences over which correct answers to favor; for example, one might prefer shorter correct answers to longer ones. We extend the framework to incorporate a preference function, as discussed in Appendix C.

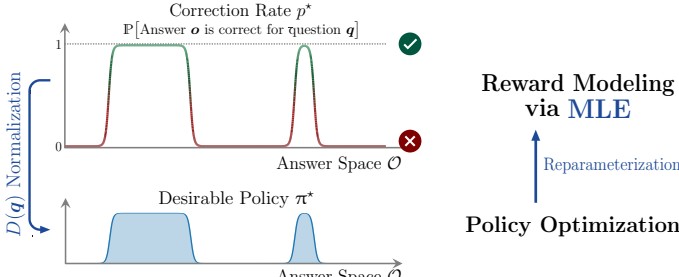

Figure 3: An optimal policy $\pi^\star$ is derived from reward probabilities $p^\star$ through normalization (see Equation (4)). This approach reframes the task of finding the best policy as a more straightforward statistical problem: learning a reward model from data.

**Maximum Likelihood Estimation (MLE) as the Learning Principle.** Formally, suppose we are given an i.i.d. dataset $\mathcal{D} = \{(\boldsymbol{q}_i, \boldsymbol{o}_i, r_i)\}_{i=1}^n$. If we have an estimate of the difficulty $D(\boldsymbol{q}_i)$ (as defined in Equation (5)), we can reparameterize the probability model as

$$p_\theta(\boldsymbol{q}, \boldsymbol{o}) = \frac{1}{D(\boldsymbol{q})} \pi_\theta(\boldsymbol{o} \mid \boldsymbol{q}), \tag{6}$$

where $\pi_\theta$ belongs to a parametric policy class. The straightforward way to solve $p_\theta$ is through the maximum likelihood (equivalently, cross-entropy minimization) objective:

$$\text{minimize}_\theta \ \ \widehat{\mathcal{L}}_0(\theta) = -\frac{1}{n} \sum_{i=1}^n \left\{ r_i \cdot \log p_\theta(\boldsymbol{q}_i, \boldsymbol{o}_i) + (1 - r_i) \cdot \log\left(1 - p_\theta(\boldsymbol{q}_i, \boldsymbol{o}_i)\right) \right\}. \tag{7}$$

Plugging in the reparameterization yields the equivalent form:

$$\text{minimize}_\theta \ \ \widehat{\mathcal{L}}(\theta) = -\frac{1}{n} \sum_{i=1}^n \left\{ r_i \cdot \log \pi_\theta(\boldsymbol{o}_i \mid \boldsymbol{q}_i) + (1 - r_i) \cdot \log\left(1 - \frac{\pi_\theta(\boldsymbol{o}_i \mid \boldsymbol{q}_i)}{D(\boldsymbol{q}_i)}\right) \right\}. \tag{8}$$

This formulation makes explicit the bridge between *policy learning* and *reward modeling*: by estimating $p^\star$, we implicitly learn a good policy $\pi_\theta$ that maximizes accuracy.

## 4.2 Calibrating Policy Gradient via MLE.

We now turn to the algorithmic perspective: how can the maximum likelihood objective (8) guide policy gradient methods? Our first step is to analyze the gradient of the MLE loss. This is summarized in Theorem 1.

**Theorem 1.** *The gradient of the log-likelihood $\widehat{\mathcal{L}}(\theta)$ with respect to the parameters $\theta$ is given by*

$$\nabla_\theta \widehat{\mathcal{L}}(\theta) = -\frac{1}{n} \sum_{i=1}^n \left\{ r_i - (1 - r_i) \frac{\pi_\theta(\boldsymbol{o}_i \mid \boldsymbol{q}_i)}{D(\boldsymbol{q}) - \pi_\theta(\boldsymbol{o}_i \mid \boldsymbol{q}_i)} \right\} \cdot \nabla_\theta \log \pi_\theta(\boldsymbol{o}_i \mid \boldsymbol{q}_i). \tag{9}$$

*Comparison with Policy Gradient.* For reference, the standard policy gradient expression for maximizing the accuracy objective in Equation (1) is

$$\nabla_\theta J(\pi_\theta) = \mathbb{E}\left[ r \cdot \nabla_\theta \log \pi_\theta(\boldsymbol{o} \mid \boldsymbol{q}) \right].$$

Classical algorithms such as REINFORCE, PPO, and GRPO are all built upon this form. In practice, the raw reward $r$ is often replaced by an advantage estimate $A$ to reduce variance. However, in GRPO, when all answers in a batch are incorrect (i.e., $r = 0$), the gradient contribution vanishes entirely (after centralization). This explains why negative groups are typically discarded in existing methods.

*MLE Perspective.* Theorem 1 sheds new light on this issue. The first term of the gradient,

$$r_i \cdot \nabla_\theta \log \pi_\theta(\boldsymbol{o}_i \mid \boldsymbol{q}_i),$$

matches the standard policy gradient signal: positive samples ($r_i = 1$) encourage the model to increase probability mass on correct answers.

But critically, the MLE gradient also contains an additional *negative sample contribution*:

$$- (1 - r_i) \, \frac{\pi_\theta(\boldsymbol{o}_i \mid \boldsymbol{q}_i)}{D(\boldsymbol{q}_i) - \pi_\theta(\boldsymbol{o}_i \mid \boldsymbol{q}_i)} \cdot \nabla_\theta \, \log \pi_\theta(\boldsymbol{o}_i \mid \boldsymbol{q}_i).$$

Although typically smaller in scale, this term is non-negligible when only negative answers are observed, or when negative samples dominate the data. In other words, discarding negative groups overlooks a legitimate part of the gradient revealed by the MLE formulation.

*Calibrated Policy Gradient.* Motivated by this observation, we propose a unified modification to REINFORCE-type algorithms for LLM reasoning. Specifically, we replace the raw reward $r = r^\star(\boldsymbol{q}, \boldsymbol{o})$ with a *calibrated reward* that incorporates both positive and negative contributions:

$$\widetilde{r} = r \; - \; (1 - r) \, \frac{\pi_\theta(\boldsymbol{o} \mid \boldsymbol{q})}{D(\boldsymbol{q}) - \pi_\theta(\boldsymbol{o} \mid \boldsymbol{q})} \, . \tag{10}$$

When the generation is correct ($r = 1$), the calibrated reward is unchanged: $\widetilde{r} = r = 1$. The adjustment applies only to incorrect samples. In negative groups, $r = 0$ for every candidate, but the policy confidences $\pi_{\theta_{\text{old}}}(\boldsymbol{o} \mid \boldsymbol{q})$ differ; consequently, the adjusted rewards $\widetilde{r}$ also differ across candidates, reflecting their relative confidence. This ensures that negative groups contribute informative gradients rather than being discarded, thereby yielding a more statistically principled update rule.

We provide the proof and show that the estimator is consistent in Appendix B.1: if the model is correctly specified (i.e., $\pi^\star = \pi_{\theta^\star} \in \{\pi_\theta\}_{\theta \in \Theta}$), then the true parameter vector $\theta^\star$ is a maximizer of the population log-likelihood.

## 4.3 CONFIDENCE WEIGHTED VALUE FUNCTION

After introducing the calibrated policy gradient, we can interpret it as solving a modified policy optimization problem with a redefined value function $J_{\text{MLE}}(\pi_\theta)$. The next theorem formalizes this perspective: in the on-policy setting, the MLE gradient coincides with the gradient of this specially constructed value function. The proof is deferred to Appendix B.2.

**Theorem 2.** *If we collect dataset $\mathcal{D}$ according to $\boldsymbol{q}_i \sim \xi$ and $\boldsymbol{o}_i \sim \pi_\theta(\cdot \mid \boldsymbol{q}_i)$, then the gradient of the (population) log-likelihood function $\mathcal{L}(\theta)$ is identical to the gradient of the following value function $J_{\text{MLE}}(\pi_\theta)$:*

$$\text{maximize}_\theta \qquad J_{\text{MLE}}(\pi_\theta) \; = \; J_+(\pi_\theta) - J_-(\pi_\theta) \, , \tag{11}$$

*where*

$$J_+(\pi_\theta) \; := \; \mathbb{E}_{\boldsymbol{q} \sim \xi, \, \boldsymbol{o} \sim \pi_\theta(\cdot \mid \boldsymbol{q})} \Big[ r^\star(\boldsymbol{q}, \boldsymbol{o}) \Big] \, , \tag{12a}$$

$$J_-(\pi_\theta) \; := \; \mathbb{E}_{\boldsymbol{q} \sim \xi, \, \boldsymbol{o} \sim \pi_\theta(\cdot \mid \boldsymbol{q})} \Big[ w\big( \pi_\theta(\boldsymbol{o} \mid \boldsymbol{q}) / D(\boldsymbol{q}) \big) \big\{ 1 - r^\star(\boldsymbol{q}, \boldsymbol{o}) \big\} \Big] \, . \tag{12b}$$

*Here the weight function $w(\cdot)$ is defined as*

$$w(z) \; := \; \frac{1}{z} \, \log \frac{1}{1 - z} - 1 \qquad \textit{for any } 0 \leq z < 1. \tag{13}$$

This formulation provides insight into the behavior of the MLE optimizer. The objective $J_{\text{MLE}}(\pi_\theta)$ balances two components:

$J_+(\pi_\theta)$**:** This is the standard policy gradient objective (REINFORCE), which maximizes the expected reward. It encourages the policy $\pi_\theta$ to take actions (i.e., propose answers $\boldsymbol{o}$) that are likely to be correct.

$J_-(\pi_\theta)$**:** This term acts as a penalty for incorrect answers. The cost of being incorrect, $1 - r^\star$, is re-weighted by $w\big( \pi_\theta(\boldsymbol{o} \mid \boldsymbol{q}) / D(\boldsymbol{q}) \big)$, which represents the policy's own "odds" of its prediction being correct. The penalty is most severe when the policy is highly confident but wrong (as $\pi_\theta \to D_-$, $w \to \infty$). Conversely, the penalty is negligible when the policy is uncertain and wrong (as $\pi_\theta \to 0_+$, $w \to 0$). It encourages diversity in the negative responses / exploration in the negative space.

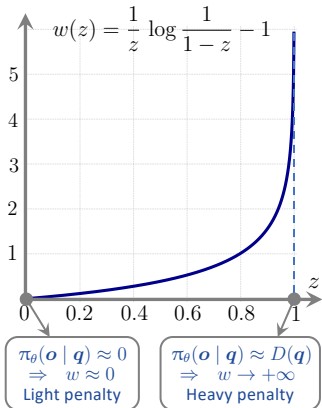

Figure 4: Illustration of the weight function $w(z)$.

The objective $J_{\mathrm{MLE}}(\pi_\theta)$ creates a powerful dynamic. It not only drives the policy to maximize rewards but, more critically, it uses the penalty term $J_-(\pi_\theta)$ to enforce "principled exploration". By penalizing misplaced confidence, the agent is forced to explore diverse responses rather than exploiting a potentially flawed understanding. This balance between exploitation and exploration is essential for learning a well-calibrated policy.

# 5 PROPOSED MODIFICATION TO GRPO

The likelihood framework naturally led to a theoretically-grounded modification to GRPO's advantage function, directly incorporating the insights from the $J_{\mathrm{MLE}}(\pi_\theta) = J_+(\pi_\theta) - J_-(\pi_\theta)$ objective to enhance exploration and policy calibration. The core of our proposal is to replace the original reward with our adjusted reward $\widetilde{r}$ from Equation (10). The adjusted reward directly implements the gradient of our theoretical objective. The calibrated reward is then normalized and the obtained advantage is used in Equation (3). We do not modify the GRPO loss function.

## 5.1 IMPLEMENTATION AND PRACTICAL CONSIDERATIONS

We calibrate rewards using the ratio $\frac{\pi_{\theta_{\mathrm{old}}}}{D(\boldsymbol{q}) - \pi_{\theta_{\mathrm{old}}}}$ which requires careful handling, particularly in how the probability $\pi_{\theta_{\mathrm{old}}}$ and the difficulty factor $D(\boldsymbol{q})$ are estimated and used.

$\pi_{\theta_{\mathrm{old}}}$ **Term.** For LLMs with long generations, raw sequence probabilities are dominated by length: per-token probabilities tend to be of similar magnitude, so the sequence probability decays roughly as $\gamma^{|\boldsymbol{o}|}$ for some $\gamma \in (0, 1)$. Consequently, plugging $\pi_{\theta_{\mathrm{old}}}$ in directly makes the adjustment sparse: length-driven decay pushes most candidates' terms to 0, while a single dominant candidate gets a much larger value. To mitigate this, we use the length-normalized (geometric-mean) probability

$$\bar{\pi}_{\theta_{\mathrm{old}}}(\boldsymbol{o} \mid \boldsymbol{q}) := \pi_{\theta_{\mathrm{old}}}(\boldsymbol{o} \mid \boldsymbol{q})^{1/|\boldsymbol{o}|}.$$

In Appendix C we show that our likelihood framework naturally generalizes to incorporate preferences over correct generations (e.g., in the example in Section 4.1, we can make $\pi^\star(A \mid \boldsymbol{q}) = \rho(\boldsymbol{q}, A)$ and $\pi^\star(B \mid \boldsymbol{q}) = \rho(\boldsymbol{q}, B)$, rather than 0.5 and 0.5; empirically, the above substitution is equivalent to a calibrated reward that encodes a length preference for correct generations.

**Estimating $D(\boldsymbol{q})$.** The true difficulty function $D(\boldsymbol{q})$ (as defined in Equation (5)) is unknown and acts as a key hyperparameter controlling learning dynamics. Smaller $D(\boldsymbol{q})$ increases the penalty on confident but incorrect predictions, encouraging broader exploration to avoid overconfidence. This mechanism allows tuning between exploiting correct answers and exploring uncertain ones.

A direct estimator follows from importance sampling:

$$D_{\mathrm{imp}}(\boldsymbol{q}) = \left\{ \sum_{\boldsymbol{o}' \in \mathcal{O}} p^\star(\boldsymbol{o}' \mid \boldsymbol{q}) \right\}^{-1} = \mathbb{E}_{\boldsymbol{o} \sim \pi_{\theta_{\mathrm{old}}}} \left[ \frac{r^\star(\boldsymbol{q}, \boldsymbol{o})}{\pi_{\theta_{\mathrm{old}}}(\boldsymbol{o} \mid \boldsymbol{q})} \right]^{-1} \approx \left\{ \frac{1}{G} \sum_{i=1}^{G} \frac{r_i}{\pi_{\theta_{\mathrm{old}}}(\boldsymbol{o}_i \mid \boldsymbol{q})} \right\}^{-1}.$$
(14)

In this formulation, we approximate the expectation with a Monte Carlo average over a group of $G$ samples $\{(\boldsymbol{o}_i, r_i)\}_{i=1}^{G}$ drawn from $\pi_{\theta_{\mathrm{old}}}$.

For numerical stability, we should conservatively *overestimate* $D(\boldsymbol{q})$ so that the denominator $D(\boldsymbol{q}) - \bar{\pi}_{\theta_{\mathrm{old}}}$ is positive and well-conditioned. Concretely, over the $G$ candidates in the group we set

$$D(\boldsymbol{q}) = \max\left( D_{\mathrm{imp}}(\boldsymbol{q}), \ 2 \cdot \max_{1 \le i \le G} \bar{\pi}_{\theta_{\mathrm{old}}}(\boldsymbol{o}_i \mid \boldsymbol{q}) \right),$$

which keeps the calibrated rewards in $[-1, 1]$.

$D_{\mathrm{imp}}(\boldsymbol{q})$ is undefined for *negative groups* as all $r_i$ are zero. In that case we fall back to

$$D(\boldsymbol{q}) = 2 \cdot \max_{1 \le i \le G} \bar{\pi}_{\theta_{\mathrm{old}}}(\boldsymbol{o}_i \mid \boldsymbol{q}).$$

**Handling Invariance.** GRPO's group-wise normalization enjoys a useful *sign invariance*: regardless of how many generations are correct, after normalization all incorrect generations have negative

advantages and all correct generations have positive advantages. We aim to preserve this property under our calibration. Consider the extreme mixed group with one correct and $G - 1$ incorrect generations; the calibrated rewards might look like $[1, 0, -1, \ldots, -1]$. To maintain sign invariance, we scale all negative calibrated rewards by $1/G$.

**Calibrated Reward (per sample).** In combination, our calibrated reward is

$$\widetilde{r}_i := r_i - (1 - r_i) \frac{1}{G} \frac{\bar{\pi}_{\theta_{\text{old}}}(\boldsymbol{o}_i \mid \boldsymbol{q})}{D(\boldsymbol{q}) - \bar{\pi}_{\theta_{\text{old}}}(\boldsymbol{o}_i \mid \boldsymbol{q})},$$

with

$$D(\boldsymbol{q}) = \begin{cases} \max\Big(D_{\text{imp}}(\boldsymbol{q}), \ 2 \cdot \max_j \bar{\pi}_{\theta_{\text{old}}}(\boldsymbol{o}_j \mid \boldsymbol{q})\Big), & \text{(mixed group)}, \\ 2 \cdot \max_j \bar{\pi}_{\theta_{\text{old}}}(\boldsymbol{o}_j \mid \boldsymbol{q}), & \text{(negative group)}. \end{cases}$$

**Final Objective.** In negative groups, the only signal comes from confidence differences rather than a verifiable reward, so we treat it as a weaker, auxiliary signal. For those groups we use de-meaning only in the normalization for simplicity, and we introduce the only hyperparameter, $\alpha$, to down-weight their contribution:

$$J_{\text{ours}} = J_{\text{GRPO}}[\text{mixed groups}] + \alpha \cdot J_{\text{GRPO}}[\text{negative groups}].$$

## 6  EXPERIMENTAL RESULTS

We now empirically test the effectiveness of our algorithm.

**Set-up.** We evaluate our method on mathematical reasoning. We conduct training on the MATH training split combined with Numina 1.5 (Li et al., 2024a). All evaluations are on the MATH test set. We consider two models, `Llama-3.1-8B-Instruct` (Dubey et al., 2024) and `Qwen-2.5-3B-Base` (Yang et al., 2024) [2], and compare our method against the baseline GRPO. To further test for generality, we also examine training on the DAPO (Yu et al., 2025a) dataset and report details and results in Appendix E.

**Training protocol.** To stress-test learning from negative groups, we use a possibly large $G$ and sample 16 completions per question. Each gradient update uses a global batch of 512 trajectories (32 questions $\times$ 16 samples). We decode with temperature 1.0 and cap generations at 4,096 tokens. We do not add any KL regularization following common practices. The negative ratio $\alpha$ is set to 0.25 for all models. No format rewards are added to the scalar reward.

**Evaluation.** At evaluation time, we use temperature 1.0 and top-$p$ 1.0 to evaluate the model in the plain setup as training, and report Pass@$k$ for $k \in \{1, 2, 4, 8, 16\}$. We present evaluation curves during training for both the full MATH dataset, and the MATH Levels 4-5 subset to understand the performance on hard questions. To test for generalization, we also include GSM8k (Cobbe et al., 2021), MinervaMath (Lewkowycz et al., 2022), and OlympiadBench (He et al., 2024) for evaluation. We use Math-Verify (Kydlíček, 2025) as the verifier function for both training and evaluation.

**Results.** We report training curves for `Llama` and `Qwen` in Figure 5. The full training results are in Appendix E. Across both models, LENS consistently attains higher accuracy than the GRPO baseline throughout training. On the hard split of MATH, LENS shows substantial additional gains, indicating that the method effectively converts *negative groups*, which often correspond to hard instances where no candidate is initially correct, into useful learning signals. As a result, when the GRPO curve saturates, LENS continues to improve. These results indicate that our method learns effectively through exploration and explicitly leverages negative groups, yielding stronger performance on difficult problems. Moreover, training remains stable for $>$1,000 steps without ad hoc tricks or collapse. Training results using DAPO training set are included in Appendix E, where we observe consistent improvements with identical hyperparameters.

We further report Pass@$k$ in Table 1. Compared with the GRPO baseline, LENS achieves higher Pass@$k$ for $k \in \{1, 2, 4, 8, 16\}$, with the improvement at Pass@16 also significant. These results

---

[2]Following prior work, we apply RL to the `Qwen` *base* model (Liu et al., 2025b), which already follows instructions and produces outputs in the required format, whereas for `Llama` we use the *instruction-tuned* model (Arnal et al., 2025). This allows us to remove the format reward in RLVR.

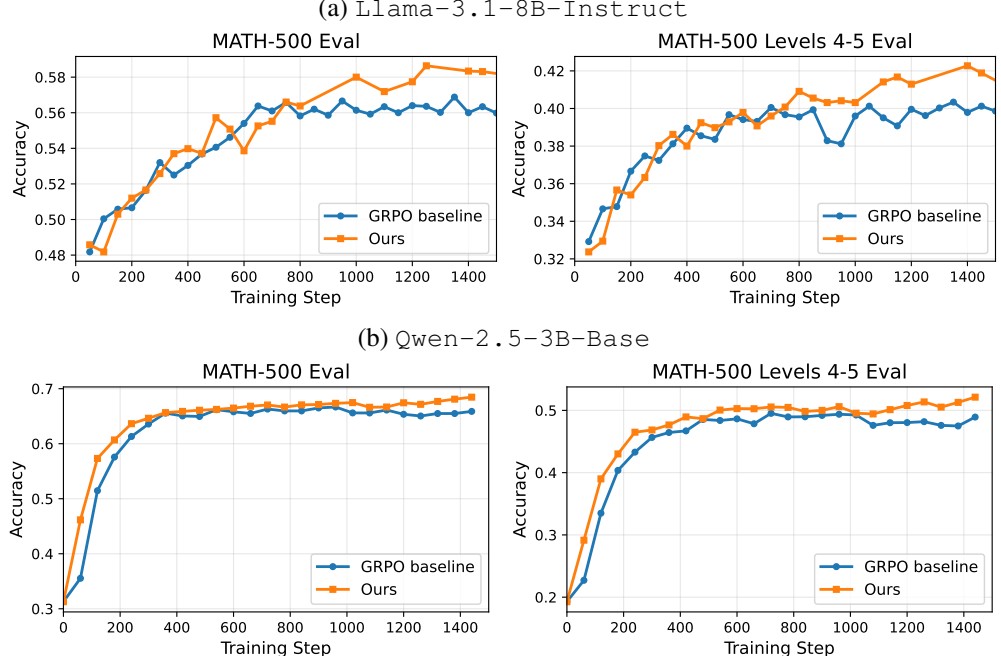

Figure 5: Comparison of our algorithm and GRPO baseline: performance on the full MATH test set and the Levels 4–5 (hard) subset. Top: `Llama-3.1-8B-Instruct`; bottom: `Qwen-2.5-3B-Base`. The accuracy is averaged over all 16 generations during the evaluation. Our algorithm brings improvement for both models.

Table 1: Pass@$k$ results on MATH with `Llama-3.1-8B-Instruct` and `Qwen-2.5-3B-Base`.

| Model | Algorithm | Pass@1 | Pass@2 | Pass@4 | Pass@8 | Pass@16 |
|---|---|---|---|---|---|---|
| Llama-3.1-8B-Instruct | GRPO baseline | 56.88 | 65.42 | 72.08 | 78.34 | 82.80 |
| | LENS (Ours) | **58.64** | **66.08** | **73.98** | **79.46** | **83.40** |
| Qwen-2.5-3B-Base | GRPO baseline | 65.88 | 72.39 | 77.82 | 82.05 | 85.13 |
| | LENS (Ours) | **68.46** | **74.74** | **79.75** | **83.54** | **86.28** |

indicate that our algorithm consistently improves Pass@$k$ for all $k$, rather than only Pass@1, and that its confidence-based design enables these exploration gains.

To verify the robustness of our findings, we conducted two independent training runs to compute the mean and standard deviation, evaluating the `Qwen` model across all five benchmarks. The results, reported in Table 2, demonstrate that: (1) our method achieves statistically significant improvements over GRPO on MATH, MATH Levels 4–5, MinervaMath, and OlympiadBench; and (2) LENS exhibits high stability with negligible deviation across seeds, when scaling RL to thousands of steps. Appendix D.2 presents ablations that separately evaluate the effect of adjusted rewards in mixed and negative groups, showing strong improvements from negative groups alone.

Table 2: Comparison of our method against the baseline using `Qwen-2.5-3B-Base`. Values denote accuracy (%) Mean $\pm$ Std. Generated with 2 random seeds.

| Evaluation Set | GRPO Baseline | LENS (Ours) |
|---|---|---|
| MATH | $66.11 \pm 0.38$ | $\mathbf{68.35 \pm 0.67}$ |
| MATH Levels 4-5 | $49.09 \pm 0.26$ | $\mathbf{51.82 \pm 0.35}$ |
| GSM8K | $85.61 \pm 0.12$ | $\mathbf{85.98 \pm 0.16}$ |
| MinervaMath | $26.67 \pm 0.45$ | $\mathbf{27.44 \pm 0.26}$ |
| OlympiadBench | $30.91 \pm 0.24$ | $\mathbf{32.78 \pm 0.27}$ |

## 7 DISCUSSION

In this paper, we start from an observation. In GRPO, any generation group in which all samples are incorrect does not contribute to the optimization, even though these generations already consume substantial compute. We ask a question: can we use this data in a principled way? We develop a theoretical framework that begins with reward modeling using both positive and negative data, connects it to policy optimization, and shows that the MLE objective corresponds to an adjusted value function. The adjustment adds a confidence-weighted penalty for incorrect generations. This view yields a calibrated reward that fits seamlessly into GRPO. Empirically, we demonstrate effectiveness on both `Llama` and `Qwen` models, with improvements across all Pass@$k$ scores.

Our empirical algorithm builds on the connection between reward modeling and policy optimization, and the framework can also incorporate preference, as shown in Appendix C. We study the simple case and leave further exploration of preference-aware variants for future work. To balance the impact of negative groups and mixed groups, we introduce a single tunable hyperparameter. A natural direction is to quantify the contributions of both sources in theory and design an objective that is free of hyperparameters. Our framework also covers nonbinary reward signals theoretically, and we defer a systematic experimental study of this setting to future work.

## ACKNOWLEDGMENT

The authors would like to sincerely thank Dulhan Jayalath, Lovish Madaan, and Yuda Song for their technical guidance. An initial part of this work was completed while YF was an intern at Meta, and YF would like to thank Cheng Zhang for hosting. YF and JK acknowledge support from the Simons Foundation through the Collaborative Grant "The Physics of Learning and Neural Computation." YD acknowledges support from NSF Grant DMS-2413812.

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

# A    OTHER RELATED WORKS

*Exploration in RL.* Enhancing exploration during RL training is an important part for all RL algorithms. In RLHF, Xie et al. (2024); Cen et al. (2024); Zhang et al. (2024) use the base model likelihood as an exploration bonus, nudging the policy toward outputs that are plausible yet seldom sampled. Closest in theoretical spirit to our view is Feng et al. (2025), which studies the MLE objective of reward modeling to derive a principled exploration method. In the reasoning setting, Gao et al. (2025) employ Random Network Distillation (Burda et al., 2018) to encourage novel solution traces. Other works (Cheng et al., 2025; Zheng et al., 2025) promote exploration through entropy based objectives. Finally, Chen et al. (2025b) optimize a pass@k objective (Tang et al., 2025) to increase batch diversity during training. However, these approaches do not propose to differentiate rewards inside negative groups and focus mainly on mixed groups.

*Asymmetric treatment of positive and negative outputs.* A few recent work introduce asymmetric treatment of positive and negative generations during REINFORCE-style training. (Roux et al., 2025) introduces an asymmetric variant of importance sampling to speed up learning. Arnal et al. (2025) demonstrate that asymmetric REINFORCE, and in particular reducing the signal from negative generations, can be beneficial when data is off-policy.

*Using Confidence in RLVR.* Confidence proxies have also been applied in RLVR, mainly proposed as a surrogate for the rule-based verifier. Zhao et al. (2025) use the KL divergence between the per token generation probability and a uniform distribution. Zhou et al. (2025); Yu et al. (2025b); Liu et al. (2025a) take the log prob of generating the reference answer conditioned on the existing CoT as the reward. Li et al. (2025) leverage confidence scores at test time for light tuning and report gains. Prabhudesai et al. (2025) similarly optimize the entropy of response tokens as the reward. In all of these studies, the rule-based reward is replaced with a confidence-based proxy and light training is performed. Most works do not train beyond one hundred steps and focus only on `Qwen` models, which raises concerns about generalization and the risk of reward hacking without a bag of tricks. In contrast, we do not aim to replace rule based rewards; instead, we propose to make use of negative groups in GRPO in a principled way. We demonstrate effectiveness on both `Llama` and `Qwen` and show stable training for more than one thousand five hundred steps.

Xu & Ding (2025) leverage on-the-fly baseline such that the negative groups will have a non-zero baseline and the advantage is not zero. Similarly, Nan et al. (2025) also employs advantage calibration to change the baseline. Le et al. (2025) leverages the entropy to create difference in the negative groups. Our work have a more theory-grounded. Xiong et al. (2025) propose to solve the negative group by adaptively allocate more generation samples for hard problems. Prakash & Buvanesh (2025) emphasize the importance to add easy sample to help generate correct answers for hard problems.

# B    PROOFS

## B.1    PROOF OF THEOREM 1

We now provide the proof of Theorem 1 and a comment on the estimator consistency.

**Proof of Theorem 1.** Let $\pi_\theta \equiv \pi_\theta(o \mid q)$ and $D \equiv D(q)$ for notational brevity. The gradient of each individual term in the loss $\widehat{\mathcal{L}}(\theta)$ with respect to $\theta$ is found using the chain rule:

$$\nabla_\theta \left[ r \cdot \log \pi_\theta + (1 - r) \cdot \log \left( 1 - \frac{\pi_\theta}{D} \right) \right] = \left( \frac{r}{\pi_\theta} - \frac{1 - r}{D - \pi_\theta} \right) \nabla_\theta \, \pi_\theta \, .$$

By applying the identity for the gradient of a logarithm, $\nabla_\theta \, \pi_\theta = \pi_\theta \cdot \nabla_\theta \log \pi_\theta$, we can express the result as:

$$\left( r - (1 - r) \frac{\pi_\theta}{D - \pi_\theta} \right) \nabla_\theta \, \log \pi_\theta \, ,$$

which provides the final result.

**Consistency of the Estimator.** A key property of this estimator is its consistency under ideal conditions. If the model is correctly specified (i.e., $\pi_{\theta^\star} \in \{\pi_\theta\}_{\theta \in \Theta}$), then the true parameter vector $\theta^\star$ is a maximizer of the population log-likelihood. This can be verified by observing that the gradient $\nabla_\theta \mathcal{L}(\theta)$ evaluates to zero at $\theta = \theta^\star$. By taking the conditional expectation of the gradient's inner term with respect to $r$, given $\boldsymbol{q}$ and $\boldsymbol{o}$, we find:

$$\mathbb{E}_{r|\boldsymbol{q},\boldsymbol{o}}\left[r - (1-r)\frac{\pi_{\theta^\star}(\boldsymbol{o} \mid \boldsymbol{q})}{D(\boldsymbol{q}) - \pi_{\theta^\star}(\boldsymbol{o} \mid \boldsymbol{q})}\right]$$

Using $\mathbb{E}[r \mid \boldsymbol{q}, \boldsymbol{o}] = p^\star(\boldsymbol{o} \mid \boldsymbol{q})$ and the definition $p^\star = \pi^\star/D$, this becomes:

$$= p^\star - (1 - p^\star)\frac{\pi_{\theta^\star}}{D - \pi_{\theta^\star}} = p^\star - (1 - p^\star)\frac{p^\star}{1 - p^\star} = p^\star - p^\star = 0 \,.$$

Since the conditional expectation of the term multiplying $\nabla_\theta \log \pi_\theta$ is zero, the full expectation is zero, confirming that $\theta^\star$ is a stationary point.

### B.2 PROOF OF THEOREM 2

We will show that $\nabla_\theta J_{\mathrm{MLE}}(\pi_\theta)$ is equivalent to $\nabla_\theta \mathcal{L}(\theta)$ when $\mu = \pi_\theta$.

First, the target gradient from Theorem 1, with the sampling policy $\mu$ set to the model policy $\pi_\theta$, is:

$$\nabla_\theta \mathcal{L}(\theta)\big|_{\mu=\pi_\theta} = \mathbb{E}_{\boldsymbol{q}\sim\xi,\,\boldsymbol{o}\sim\pi_\theta(\cdot|\boldsymbol{q})}\left[\left\{r - (1-r)\frac{\pi_\theta}{D - \pi_\theta}\right\} \cdot \nabla_\theta \log \pi_\theta(\boldsymbol{o} \mid \boldsymbol{q})\right]. \quad (15)$$

Next, we rigorously compute the gradient of $J(\pi_\theta) = J_+(\pi_\theta) - J_-(\pi_\theta)$. The gradient of the positive term is standard:

$$\nabla_\theta J_+(\pi_\theta) = \mathbb{E}_{\boldsymbol{q}\sim\xi,\,\boldsymbol{o}\sim\pi_\theta(\cdot|\boldsymbol{q})}\left[r \cdot \nabla_\theta \log \pi_\theta\right]. \quad (16)$$

For the negative term, $J_-(\pi_\theta) = \mathbb{E}_{\boldsymbol{o}\sim\pi_\theta}\left[w(\pi_\theta/D) \cdot (1-r)\right]$, we use the product rule and derive

$$\nabla_\theta J_-(\pi_\theta) = \mathbb{E}_{\boldsymbol{q},\boldsymbol{o}\sim\pi_\theta}\left[(1-r)\big(w(\pi_\theta/D) + (\pi_\theta/D) \cdot w'(\pi_\theta/D)\big) \cdot \nabla_\theta \log \pi_\theta\right]. \quad (17)$$

Now we compute $w(z) + z \cdot w'(z)$:

$$w(z) + z \cdot w'(z) = \left(\frac{-\log(1-z)}{z} - 1\right) + z\left(\frac{\frac{z}{1-z} + D\log(1-z)}{z^2}\right) = \frac{1}{1-z} - 1 = \frac{z}{1-z} \,.$$

This is exactly the term we needed. Substituting this result back into the gradient for $J_-(\pi_\theta)$:

$$\nabla_\theta J_-(\pi_\theta) = \mathbb{E}_{\boldsymbol{q},,\boldsymbol{o}\sim\pi_\theta}\left[(1-r)\left(\frac{\pi_\theta}{D - \pi_\theta}\right) \cdot \nabla_\theta \log \pi_\theta\right]. \quad (18)$$

Finally, combining the gradients for the positive and negative parts of $J(\pi_\theta)$:

$$\nabla_\theta J_{\mathrm{MLE}}(\pi_\theta) = \nabla_\theta J_+(\pi_\theta) - \nabla_\theta J_-(\pi_\theta) \quad = \mathbb{E}_{\boldsymbol{q},,\boldsymbol{o}\sim\pi_\theta}\left[\left(r - (1-r)\frac{\pi_\theta}{D - \pi_\theta}\right) \cdot \nabla_\theta \log \pi_\theta\right]. \quad (19)$$

This expression is identical to the MLE gradient in equation 15. The equivalence is proven.

## C A PREFERENCE-AWARE FRAMEWORK

The framework introduced in Section 4.1 assumed that when multiple answers are correct, the optimal policy distributes probability mass uniformly across them. For example, if both $A$ and $B$ are correct answers to a question $\boldsymbol{q}$, we had $\pi^\star(A \mid \boldsymbol{q}) = \pi^\star(B \mid \boldsymbol{q}) = 0.5$. However, uniformity may not always reflect the true reasoning process. In practice, we might prefer some answers over others. For instance, $A$ could be easier to infer, shorter in form, or more natural to express. In such cases, a more realistic distribution might be $\pi^\star(A \mid \boldsymbol{q}) = 0.9$ and $\pi^\star(B \mid \boldsymbol{q}) = 0.1$.

From the perspective of chain-of-thought reasoning, preferences can capture properties such as the length of the reasoning trajectory or the similarity of an answer to outputs from a reference language model. To encode this flexibility, we introduce a nonnegative *preference function*:

$$\rho(\boldsymbol{q}, \boldsymbol{o}) \geq 0,$$

which adjusts the weight assigned to each $(\boldsymbol{q}, \boldsymbol{o})$ pair.

**Modified Framework.** With the preference function, we adjust the relation between policy $\pi_\theta$ and correctness probabilities. Specifically, we define

$$p_\theta(\boldsymbol{q}, \boldsymbol{o}) = \frac{1}{D(\boldsymbol{q}) \cdot \rho(\boldsymbol{q}, \boldsymbol{o})} \, \pi_\theta(\boldsymbol{o} \mid \boldsymbol{q}), \tag{20}$$

where the difficulty factor $D(\boldsymbol{q})$ is updated as

$$D(\boldsymbol{q}) = \left\{ \sum_{\boldsymbol{o} \in \mathcal{O}} p^\star(\boldsymbol{q}, \boldsymbol{o}) \cdot \rho(\boldsymbol{q}, \boldsymbol{o}) \right\}^{-1}. \tag{21}$$

Intuitively, $D(\boldsymbol{q})$ still measures how hard the question is, but it now accounts for the internal weighting across candidate answers.

The maximum likelihood estimation (MLE) problem under this new framework becomes

$$\min_\theta \quad \widehat{\mathcal{L}}(\theta) = -\frac{1}{n} \sum_{i=1}^{n} \left\{ r_i \cdot \log \pi_\theta(\boldsymbol{o}_i \mid \boldsymbol{q}_i) + (1 - r_i) \cdot \log \left( 1 - \frac{\pi_\theta(\boldsymbol{o}_i \mid \boldsymbol{q}_i)}{D(\boldsymbol{q}_i) \cdot \rho(\boldsymbol{q}_i, \boldsymbol{o}_i)} \right) \right\}. \tag{22}$$

The corresponding gradient of the log-likelihood is

$$\nabla_\theta \widehat{\mathcal{L}}(\theta) = -\frac{1}{n} \sum_{i=1}^{n} \left\{ r_i - (1 - r_i) \frac{\pi_\theta(\boldsymbol{o}_i \mid \boldsymbol{q}_i)}{D(\boldsymbol{q}_i) \cdot \rho(\boldsymbol{q}_i, \boldsymbol{o}_i) - \pi_\theta(\boldsymbol{o}_i \mid \boldsymbol{q}_i)} \right\} \cdot \nabla_\theta \log \pi_\theta(\boldsymbol{o}_i \mid \boldsymbol{q}_i). \tag{23}$$

Compared to the uniform case, the gradient now incorporates the additional signal encoded by $\rho$, ensuring that both positive and negative samples are scaled according to the chosen preference structure.

**Examples of Preference Functions.** To illustrate the flexibility of this framework, we describe some concrete choices of $\rho$:

*Preference as the data collection distribution.* Suppose we take $\rho(\boldsymbol{q}, \boldsymbol{o}) = \mu(\boldsymbol{o} \mid \boldsymbol{q})$, where $\mu$ is the distribution used to collect the dataset $\mathcal{D}$. Then the difficulty factor $D(\boldsymbol{q})$ can be approximated by:

$$D(\boldsymbol{q}) \approx \left\{ \frac{1}{|\mathcal{O}_\mathcal{D}(\boldsymbol{q})|} \sum_{\boldsymbol{o} \in \mathcal{O}_\mathcal{D}(\boldsymbol{q})} r^\star(\boldsymbol{q}, \boldsymbol{o}) \right\}^{-1},$$

where $\mathcal{O}_\mathcal{D}(\boldsymbol{q})$ denotes the set of observed answers to question $\boldsymbol{q}$ in $\mathcal{D}$. In words, $D(\boldsymbol{q})$ can be estimated as the inverse of the empirical correctness rate.

*Preference as the policy itself.* If we further set $\mu = \pi_\theta$, then the negative calibration term simplifies to

$$\frac{\pi_\theta(\boldsymbol{o}_i \mid \boldsymbol{q}_i)}{D(\boldsymbol{q}_i) \cdot \rho(\boldsymbol{q}_i, \boldsymbol{o}_i) - \pi_\theta(\boldsymbol{o}_i \mid \boldsymbol{q}_i)} = \frac{1}{D(\boldsymbol{q}_i) - 1}.$$

In this case, the weight for negative samples is exactly the correction rate of the current policy $\pi_\theta$. Equivalently, in the ordinary policy gradient formulation, each question should be reweighted by its correction rate. Although this choice does not produce the "confidence-based" weighting we ultimately seek, it highlights that commonly used uniform weights (e.g., Arnal et al. (2025); Zhu et al. (2025)) emerge as a special case of our framework.

*Preference as a function of response length.* Now, consider a preference function that depends on the length of the candidate answer:

$$\rho(\boldsymbol{q}, \boldsymbol{o}) := \gamma^{|\boldsymbol{o}|} \qquad \text{for a fixed parameter } \gamma \in (0, 1).$$

Define the shorthand

$$\bar{\pi}_\theta(\boldsymbol{o} \mid \boldsymbol{q}) := \pi_\theta(\boldsymbol{o} \mid \boldsymbol{q})^{\frac{1}{|\boldsymbol{o}|}}.$$

The negative-sample reward can then be expressed as

$$\widetilde{r}_\theta(\boldsymbol{o} \mid \boldsymbol{q}) \;=\; -\frac{\pi_\theta(\boldsymbol{o} \mid \boldsymbol{q})}{D(\boldsymbol{q}) \cdot \rho(\boldsymbol{q}, \boldsymbol{o}) - \pi_\theta(\boldsymbol{o} \mid \boldsymbol{q})} \;=\; -\frac{\bar{\pi}_\theta(\boldsymbol{o} \mid \boldsymbol{q})^{|\boldsymbol{o}|}}{D(\boldsymbol{q}) \cdot \gamma^{|\boldsymbol{o}|} - \bar{\pi}_\theta(\boldsymbol{o} \mid \boldsymbol{q})^{|\boldsymbol{o}|}} \;.$$

For large $|\boldsymbol{o}|$, we have $D(\boldsymbol{q})^{\frac{1}{|\boldsymbol{o}|}} \approx 1$. If $\gamma$ is chosen on the same scale as $\bar{\pi}_\theta$, this weight simplifies to

$$\widetilde{r}_\theta(\boldsymbol{o} \mid \boldsymbol{q}) \;=\; -\left\{ \left( \frac{D(\boldsymbol{q})^{\frac{1}{|\boldsymbol{o}|}} \cdot \gamma}{\bar{\pi}_\theta(\boldsymbol{o} \mid \boldsymbol{q})} \right)^{|\boldsymbol{o}|} - 1 \right\}^{-1} \;\approx\; -\frac{1}{|\boldsymbol{o}|} \left\{ \frac{D(\boldsymbol{q})^{\frac{1}{|\boldsymbol{o}|}} \cdot \gamma}{\bar{\pi}_\theta(\boldsymbol{o} \mid \boldsymbol{q})} - 1 \right\}^{-1}$$

$$=\; -\frac{1}{|\boldsymbol{o}|} \cdot \frac{\bar{\pi}_\theta(\boldsymbol{o} \mid \boldsymbol{q})}{D(\boldsymbol{q})^{\frac{1}{|\boldsymbol{o}|}} \cdot \gamma - \bar{\pi}_\theta(\boldsymbol{o} \mid \boldsymbol{q})} \;\approx\; -\frac{1}{|\boldsymbol{o}|} \cdot \frac{\bar{\pi}_\theta(\boldsymbol{o} \mid \boldsymbol{q})}{\gamma - \bar{\pi}_\theta(\boldsymbol{o} \mid \boldsymbol{q})} \;.$$

Therefore, in practice, it is convenient to set negative-sample reward

$$\widetilde{r}_\theta(\boldsymbol{o} \mid \boldsymbol{q}) \;:=\; -\frac{1}{|\boldsymbol{o}|} \cdot \frac{\bar{\pi}_\theta(\boldsymbol{o} \mid \boldsymbol{q})}{\gamma - \bar{\pi}_\theta(\boldsymbol{o} \mid \boldsymbol{q})} \;=\; -\frac{1}{|\boldsymbol{o}|} \cdot \frac{\pi_\theta(\boldsymbol{o} \mid \boldsymbol{q})^{\frac{1}{|\boldsymbol{o}|}}}{\gamma - \pi_\theta(\boldsymbol{o} \mid \boldsymbol{q})^{\frac{1}{|\boldsymbol{o}|}}}$$

with $\gamma > 0$ properly tuned.

# D  EXPERIMENT DETAILS

## D.1  HYPERPARAMETERS

We use a learning rate $3e - 7$ for `Llama-3.1-8B-Instruct` and a learning rate $1e - 6$ for `Qwen-2.5-3B-Base`. The base model requires a larger learning rate while the instruct model has gone through the RLHF stages so a smaller learning rate is better. Prior works (Zhu et al., 2025; Arnal et al., 2025) have used the same setup. The batch size is set to be 512, with 32 questions and 16 generations for each. We use a clipping ratio of 0.2 for all the models to mitigate the impact of off-policy data. We set the maximum number of off-policy updates to 4; in VeRL (Sheng et al., 2024), this is implemented by using a training batch size as 128 (4×32).

We set temperature and top-p to 1.0 during both training and evaluation for both models.

## D.2  ABLATION

We also conduct an ablation to understand where the improvement comes from. In our algorithm, we modify the reward for all incorrect generations in both mixed and negative groups as in Equation 10. Compared with GRPO, we adjust rewards for incorrect generations within mixed groups, and negative groups now have nonzero advantages. To quantify the contribution of each component, we use the `Llama` model and consider two settings: (i) modify only the incorrect generations in mixed groups while keeping advantages for negative groups at zero, and (ii) modify only the incorrect generations in negative groups while leaving mixed groups unchanged. This design isolates the effect of each part. We refer to these variants as *LENS with only mixed groups* and *LENS with only negative groups*. The training set is MATH and Numina 1.5. The pass@k results are reported in Table 3.

Table 3: Ablation results of pass@$k$ on MATH with `Llama-3.1-8B-Instruct`.

| Algorithm | Pass@1 | Pass@2 | Pass@4 | Pass@8 | Pass@16 |
|---|---|---|---|---|---|
| GRPO baseline | 56.88 | 65.42 | 72.08 | 78.34 | 82.80 |
| LENS with only mixed groups | 57.42 | 65.82 | 73.08 | 78.80 | 83.20 |
| LENS with only negative groups | 58.14 | **66.48** | 73.46 | **79.79** | **83.40** |
| LENS (Ours) | **58.64** | 66.08 | **73.98** | 79.46 | **83.40** |

The results show that both components help improve performance. Specifically, adjusting the reward in mixed groups encourages exploration in batches that already contain a correct answer. This helps the model reinforce correct samples while rejecting incorrect generations. As a result, *LENS with only mixed groups* yields its largest gains at pass@1. *LENS with only negative groups* also improves over GRPO and in several cases nearly matches the full *LENS*, suggesting that a substantial portion of the improvement arises from the negative groups.

# E  ADDITIONAL RESULTS

We report additional results from two training setups using distinct corpora: (i) MATH + Numina 1.5 and (ii) DAPO. These complementary results, omitted from the main paper for space, are summarized as follows. Figure 6 shows training curves for `Llama` trained on DAPO and `Qwen` trained on MATH and Numina 1.5. Table 4 reports the Pass@$k$ results for the DAPO-trained models. On this training set, we significantly improve Pass@$k$ for larger $k$, indicating greater diversity.

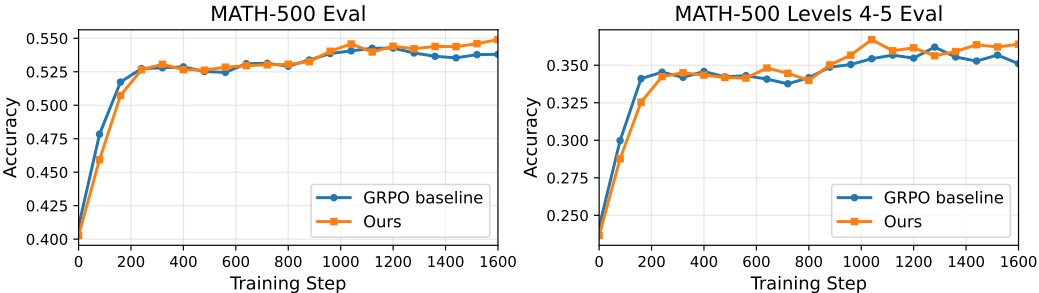

Figure 6: Comparison of our algorithm and GRPO baseline on MATH, during training: performance on the full test set and the Levels 4–5 (hard) subset. `Llama-3.1-8B-Instruct` trained on DAPO. The accuracy is averaged over all 16 generations during the evaluation. Our algorithm brings significant improvement for both models.

Table 4: Pass@$k$ results on MATH with `Llama-3.1-8B-Instruct`. Training set: DAPO.

| Model | Algorithm | Pass@1 | Pass@2 | Pass@4 | Pass@8 | Pass@16 |
|---|---|---|---|---|---|---|
| Llama-3.1-8B-Instruct | GRPO baseline | 53.80 | 61.04 | 67.30 | 71.36 | 74.54 |
| | LENS (Ours) | **54.90** | **63.03** | **69.47** | **74.36** | **77.95** |

