# OpenReview forum: "Don’t Waste Mistakes: Leveraging Negative RL-Groups via Confidence Reweighting"
_ICLR.cc/2026/Conference — Submitted to ICLR 2026_

### Official Review · Reviewer_y75z · 2025-10-18

**Soundness:** 3
**Presentation:** 3
**Contribution:** 2
**Rating:** 4
**Confidence:** 4

**Summary:**

This paper addresses sample inefficiencies in RLVR for reasoning tasks, particularly the GRPO framework. In GRPO, all positive and all negative groups yield zero gradients and thus waste compute.

The authors propose LENS (Likelihood Estimation with Negative Samples), which connects MLE in reward modeling with policy gradient methods. By reinterpreting MLE gradients, they derive a confidence-weighted penalty for incorrect generations: the more confident the model is in a wrong answer, the larger the penalty. This converts previously uninformative negative groups into useful training signals.

LENS is implemented as a drop-in modification to GRPO, introducing nonzero, confidence-dependent rewards for incorrect responses with minimal computational overhead. Experiments on MATH with LLaMA-3.1-8B and Qwen-2.5-3B show consistent performance gains across all Pass@k metrics, especially on harder problems (Levels 4–5).

Overall, the paper is good on the theoretical side and could be further improved on the experimental side. I hence evaluate it as a borderline reject at this stage. I would be happy to increase my evaluation if the experimental side is improved in the next version.

**Strengths:**

1. **Clear motivation and practical relevance.** The inefficiency of negative groups in GRPO is a well-recognized issue, and LENS directly tackles this without additional supervision.

2. **Theoretical derivation connecting MLE and policy gradients.** The reparameterization and gradient equivalence analysis (Eq. 8–10, Theorem 1) are elegant and make the proposed modification principled rather than heuristic.

3. **Simple and easily adoptable algorithm.** The proposed confidence-based correction term integrates naturally into existing GRPO setups (and potentially, many other variants), making it practical for real-world RLVR pipelines.

4. **The paper is well written and easy to follow.** Figures (especially Fig. 1 and Fig. 4) effectively illustrate how negative samples are “rescued” and how the modified rewards reshape the learning dynamics.

**Weaknesses:**

1. **Limited empirical scope.**
Experiments are restricted to MATH reasoning; no evidence is given for generalization to other RLVR tasks (e.g., code or symbolic reasoning) or at least, on other commonly used math datasets (e.g., AIME, AMC, Minerva, etc.). The claim of generality would be stronger with broader benchmarks.

2. **Absence of ablation and sensitivity studies, simple baseline.**
It’s unclear how sensitive performance is to the choice of alpha. In other words, do the benefits mainly come from the mixed group or the negative groups? Also, no comparison is made to simple baselines that also take advantage of the negative examples. (The paper mentioned some, but does not compare them.)

**Questions:**

1. Is the proposed reward shaping applied at the response level or the token level? In other words, do all tokens within a single response share the same effective learning signal (or equivalent learning rate), or is the penalty distributed across tokens differently?
This distinction is important because prior work [1] has shown that in a “negative” response, most intermediate reasoning tokens may still be correct, with only the final token being wrong. Penalizing the entire sequence uniformly might therefore suppress useful intermediate reasoning behaviors. Could the authors clarify how LENS handles this situation, specifically, whether it can assign token-wise negative gradients more selectively or whether the entire response receives a uniform penalty?

2. LENS takes good use of the all-negative groups, what about the all-positive group? Can the methodbe  extended to this case?

[1] Deng, Wenlong, et al. "On the Effect of Negative Gradient in Group Relative Deep Reinforcement Optimization." *arXiv preprint arXiv:2505.18830* (2025).

---

> ### Author Response · Authors · 2025-11-25
>
> We thank the reviewer for their careful review and insightful questions. We are particularly encouraged that you found our theoretical results strong and easy to follow. We address the specific concerns below.
>
> > Limited empirical scope
>
> Please see General Response 1.
>
> > Absence of ablation and sensitivity studies, simple baseline. It’s unclear how sensitive performance is to the choice of alpha. In other words, do the benefits mainly come from the mixed group or the negative groups? Also, no comparison is made to simple baselines that also take advantage of the negative examples.
>
> Comparison with Simple Baselines: Please refer to General Response 3.
>
> Source of Benefits: Appendix Table 1 explicitly isolates the contribution of each component. The results demonstrate that the gains are not driven by a single factor; rather, both the modification for mixed groups and for negative groups are necessary to achieve the reported performance.
>
> Sensitivity to $\alpha$: We have conducted a sensitivity analysis for $\alpha$ in the revised manuscript. We observe that performance remains robust with $\alpha \in [0.25, 0.5]$ achieving accuracies of 68.46% and 68.96%, respectively, for Qwen-2.5-3B-Base on MATH. Both results are significantly higher than the baseline, confirming that the method is not overly sensitive to precise hyperparameter tuning.
>
> > Is the proposed reward shaping applied at the response level or the token level? In other words, do all tokens within a single response share the same effective learning signal (or equivalent learning rate), or is the penalty distributed across tokens differently? This distinction is important because prior work [1] has shown that in a “negative” response, most intermediate reasoning tokens may still be correct, with only the final token being wrong. Penalizing the entire sequence uniformly might therefore suppress useful intermediate reasoning behaviors. Could the authors clarify how LENS handles this situation, specifically, whether it can assign token-wise negative gradients more selectively or whether the entire response receives a uniform penalty?
>
> Thanks for pointing out this nice connection. We clarify that LENS operates at the response level. The reward signal computed in Eq. (10) is applied uniformly to all tokens within a trajectory.
>
> Addressing the concern raised by prior work [1] requires Process Reward Models (PRMs) or token-level credit assignment, which necessitate dense supervision or complex heuristics that are challenging.
>
> Instead of attempting to surgically penalize specific tokens (which is fundamentally hard without strong process judge), LENS addresses the issue by encouraging exploration. By specifically penalizing confident incorrect solutions, LENS prevents the model from collapsing onto a partially correct but ultimately wrong path, forcing it to explore alternative reasoning trajectories.
>
> Crucially, we emphasize that our method extracts rich learning signals using only sparse outcome-based rewards, thereby avoiding the substantial annotation and computational overhead of process supervision.
>
> While deriving a token-level reward function from our MLE perspective is theoretically intriguing, it would require decomposing the partition function at the token level—a significant mathematical extension that we leave for future work.
>
>
> > LENS takes good use of the all-negative groups, what about the all-positive group? Can the methodbe extended to this case?
>
> This is an insightful suggestion. In our current framework, we assume a uniform distribution for the target policy over correct solutions, when constructing from the reward modeling as in Eq. 4. This implies that all correct generations are treated as equally desirable; consequently, an "all-positive" group offers no contrastive signal for optimization.
>
> However, the method can indeed be extended. As outlined in Appendix C, we can adopt a preference-aware framework. For instance, we can prefer shorter correct responses over longer ones. In such a case, the model could leverage all-positive groups by having a length-dependent reward design. The primary challenge lies in designing robust priors. As this requires substantial investigation, we leave it as a promising direction for future work.
>
> We thank the reviewer once again for the time and effort. Please let us know if we have addressed all the questions.

---

### Official Review · Reviewer_bfBj · 2025-10-30

**Soundness:** 3
**Presentation:** 2
**Contribution:** 2
**Rating:** 4
**Confidence:** 4

**Summary:**

The paper proposes LENS, a modification to GRPO that introduces a confidence-dependent penalty for incorrect responses derived from an MLE reformulation. The authors argue that GRPO “wastes” negative groups (groups where all responses are incorrect) because their advantages collapse to zero. LENS uses the model's predicted probabilities to build a calibrated reward that assigns non-zero penalties for wrong answers.

**Strengths:**

1. Addresses a real inefficiency in GRPO: lack of signal in all-negative groups.

2. Attempts to reframe reward modeling and RLVR through an MLE lens.

**Weaknesses:**

## Weaknesses

1. **Limited Experimental Scope.**
   The empirical evaluation covers only the MATH dataset and two model sizes. This limited scope is insufficient to support claims of generality or broad applicability across reasoning tasks or model scales.

2. **Unrealistic Assumptions About the Output Space.**
   The theoretical formulation assumes a finite, enumerable answer space. In LLM reasoning, however, the number of textual realizations of the same correct answer (e.g., *“The answer is 42.”*, *“It is 42.”*, *“42.”*, etc.) is effectively unbounded. Consequently, \(D(q)\) may be infinite or undefined, making the proposed reparameterization inapplicable to real generative settings.

3. **Breakdown of the MLE/IS Derivation for Negative Groups.**
   I was enjoying the MLE and importance-sampling derivation, but the analysis breaks down abruptly for negative groups (where all \( r_i = 0 \)). Instead of providing a principled extension of the theory for difficulty measure (such as mixed group), the paper introduces an ad-hoc fallback
   \[
   D(q) = 2 \cdot \max_i \pi_{\text{old}}(o_i \mid q),
   \]
   which has no theoretical justification. This weakens the claimed theoretical connection between reward modeling and policy optimization, especially in emphasizing resolving the learning signal from the incorrect group.

4. **Only Marginal Gains in Mixed Groups.**
   Appendix Table 2 shows that LENS provides only minor improvements on mixed groups. This suggests that most of the observed empirical gains stem from the heuristic handling of all-negative groups, rather than from the theoretically grounded likelihood-based calibration intended for mixed groups, raising a concern about the validity of the theoretical framework.

5. **Missing Comparisons to Strong Negative-Reward Baselines.**

   The paper does not compare LENS against simpler, well-established negative reinforcement strategies (e.g., penalizing all incorrect responses in all-negative groups). Prior work [1] demonstrates the strength of this method. Without these comparisons, it is unclear whether LENS provides meaningful advantages over existing methods.

[1] The Surprising Effectiveness of Negative Reinforcement in LLM Reasoning,

**Questions:**

See weakness

---

> ### Author Response · Authors · 2025-11-25
>
> We thank the reviewer for the time and effort invested in this in-depth review, and for appreciating our theoretical perspective. We address the specific concerns below.
>
> > Limited Experimental Scope. The empirical evaluation covers only the MATH dataset and two model sizes.
>
> Please see General Response 1. We add error bars and include evaluation on three more datasets.
>
> > Unrealistic Assumptions About the Output Space.
>
> Thank you for raising this point. Our theoretical exposition begins with the simplifying assumption of a finite, enumerable answer space because it makes the core ideas—particularly the reparameterization in equations (4)–(8)—easy to present in a clean algebraic form. However, none of our results rely on finiteness. The framework already extends directly to general measurable output spaces.
>
> In fact, Appendix C develops the general setting. There we assume a nonnegative measure (or preference function) $\rho(q,\cdot)$ over a measurable output space $\mathcal O$; $\rho$ does not need to normalize to $1$. In this formulation, the normalization term becomes
> $D(q)=(\int_{\mathcal O} p^\star(q,o)\,\rho(q,do))^{-1}$,
> which is well defined because $p^\star(q,o)\in[0,1]$ is uniformly bounded. When $\mathcal O$ is finite and $\rho$ is discrete and uniform, this reduces exactly to the summation form used in the main text.
>
> Under this measure-theoretic formulation, the calibrated negative penalty takes the form
> $\frac{\pi_\theta(do\mid q)}{D(q)\,\rho(q,do)-\pi_\theta(do\mid q)}$,
> which collapses to equation (23) when specialized to the finite case. Therefore, the entire construction already applies to general measurable output spaces; the finite setting in the main text is purely for readability, not a limitation of the method.
>
>
> > Breakdown of the MLE/IS Derivation for Negative Groups.
>
> Please see General Response 2.
>
> > Only Marginal Gains in Mixed Groups. Appendix Table 2 shows that LENS provides only minor improvements on mixed groups. This suggests that most of the observed empirical gains stem from the heuristic handling of all-negative groups, rather than from the theoretically grounded likelihood-based calibration intended for mixed groups, raising a concern about the validity of the theoretical framework.
>
> The difference between mixed groups and negative groups is only about the Importance Sampling derivation. We are still leverating the general theoretical framework. Appendix Table 2 shows that in the final receipe, negative groups contribute more than the mixed groups, to compare with the baseline. This is intuition, as mixed groups already have a strong signal from the correct generation.
>
> This does not show that mixed groups cannot bring better performance, as some of the scaling is designed to unify all the reward modification. There could be a design that apply only to mixed groups and achieve higher performance (should we add this?)
>
> > Missing Comparisons to Strong Negative-Reward Baselines.
>
> Please see General Response 3.
>
> We thank the reviewer once again for the time and effort. Please let us know if we have addressed all the questions.

---

### Official Review · Reviewer_285B · 2025-11-02

**Soundness:** 3
**Presentation:** 2
**Contribution:** 2
**Rating:** 4
**Confidence:** 4

**Summary:**

The paper targets RLVR training with GRPO, arguing that “negative groups” (all sampled answers wrong) are wasted because GRPO assigns them zero advantage. The authors derive a calibrated policy gradient whose reward for incorrect samples is down-weighted by model confidence and an instance-level “difficulty” term. Their modified reward yields non-zero, confidence-dependent signals for wrong answers and they implement can simply swap GRPO’s reward, using length-normalized probabilities and a simple estimator for difficulty term. They claim negligible overhead and better Pass@k on MATH with Llama-3.1-8B and Qwen-2.5-3B, especially on harder problems.

**Strengths:**

- The paper identifies a intuitive and significant inefficiency in a widely used algorithm. The concept of "wasted mistakes" is clearly state and the paper provides strong, data-driven motivation in Figure 2. 1 is an excellent framing of the problem.

- The authors provide theoretical insights to conceptually bridge from a reward modeling MLE objective to a policy gradient form, and the theoretical results provides a solid starting point for the subsequent algorithmic development.

**Weaknesses:**

- While the theoretical results are interesting, their connection to the proposed method feels weak. Some aspects of the design appear to be heuristic and lack clear justification (see details in the questions).

- The evaluation overlooks several important baselines, and some claims made in the paper are not adequately supported by evidence (see details in the questions).

**Questions:**

- Concurrent work (Xiong et al. 2025, arXiv:2504.11343) provides empirical evidence that discarding negative groups is GRPO's main advantage. Can the authors comment on this fundamental contradiction and justify why their premise holds in light of this conflicting evidence?

- The key difference between LENS and cited work (Zhu et al. 2025) is the stratification of negative rewards (confidence-weighted vs. uniform). Why was a uniform negative reward baseline (i.e., NSR from Zhu et al.) omitted from the ablation study in Table 2?

- Section 5 introduces at least four major heuristics (length-norm, a complex $D(q)$ estimator, $1/G$ scaling, and an $\alpha=0.25$ hyperparameter) that do not appear in the " derivation in Section 4. Does the theoretical reward from Eq. 10 work in practice? If not, doesn't this imply the theoretical framework is a loose inspiration rather than a principled derivation?

- The paper claims “negligible computational overhead”. Can the authors please provide quantitative data?

- Can the authors provide a theoretical justification for the number 2 in the $D(q)$ estimator 1 (line 370) and explain why the estimator must be defined differently for mixed vs. negative groups?

---

> ### Author Response · Authors · 2025-11-25
>
> We thank the reviewer for their effort in carefully reviewing our work, and we are pleased that you consider our results a solid starting point for the field. We address your questions and concerns below.
>
> > Concurrent work (Xiong et al. 2025, arXiv:2504.11343) provides empirical evidence that discarding negative groups is GRPO's main advantage. Can the authors comment on this fundamental contradiction and justify why their premise holds in light of this conflicting evidence?
>
> Thanks for raising this point. There is no contradiction. Xiong et al. (2025) show that under naive $\pm 1$ rewards, negative groups yield high-variance, misleading gradients, so discarding them (as GRPO implicitly does) is better than using them poorly. We fully agree.
>
> LENS asks a different question: can negative groups be reweighted so that they become informative rather than harmful? Crucially, our rewards are no longer uniform −1; the MLE-derived calibration reshapes these signals according to confidence, turning exactly the “noise” Xiong et al. identify into structured, informative penalties.
>
> Empirically, LENS outperforms GRPO—even though GRPO represents the best “discarding” strategy—showing that properly calibrated negative groups contain value that discarding cannot access.
>
> Thus: Xiong et al. $\rightarrow$ naive negative updates are harmful; LENS $\rightarrow$ calibrated negative updates unlock strictly better learning. Their findings do not contradict ours—they motivate the need for LENS.
>
> > The key difference between LENS and cited work (Zhu et al. 2025) is the stratification of negative rewards (confidence-weighted vs. uniform). Why was a uniform negative reward baseline (i.e., NSR from Zhu et al.) omitted from the ablation study in Table 2?
>
> Please see General Response 3.
>
> > Section 5 introduces at least four major heuristics (length-norm, a complex
>  estimator $D(q)$, $1/G$ scaling, and an $\alpha=0.25$ hyperparameter) that do not appear in the " derivation in Section 4. Does the theoretical reward from Eq. 10 work in practice? If not, doesn't this imply the theoretical framework is a loose inspiration rather than a principled derivation?
>
> We thank the reviewer for this comment. Please refer to General Response 2 regarding the estimator $D(q)$. As argued there, the use of approximations does not imply the framework is a "loose inspiration." Rather, it follows standard machine learning practice (analogous to Ridge Regression) where the functional form of the objective is derived from theory, but heurisitcs on regularization hyperparameters are required for empirical stability.
>
> Estimator $D(q)$: As detailed in GR 2, this acts as a proxy for the inverse partition function. It is necessary because the theoretical estimator is undefined in all-negative groups, serving as a design balancing stability and informativeness.
>
> Length-Norm: This is not ad-hoc but a principled derivation. As shown in Appendix C (lines 351–354), it corresponds to applying a length-aware prior to the policy in Eqn (4).
>
> $1/G$ Scaling: It is critical for preserving advantage sign invariance in GRPO; without it, incorrect samples in mixed groups could inadvertently receive positive updates due to unscaled baseline subtraction.
>
> $\alpha$: It balances exploration and exploitation, specifically modulating the trade-off between maximizing rewards in mixed groups and enforcing constraints in negative groups.
>
> Therefore, the theoretical framework still holds; Eq. (10) remains the fundamental driver of the learning signal, while these components ensure the optimization remains numerically stable.
>
> > The paper claims “negligible computational overhead”. Can the authors please provide quantitative data?
>
> The computational overhead is minimal because the reward adjustment relies on simple algebraic operations rather than additional model forward passes. Quantitatively, we observed that the runtime of LENS for the first 800 steps is comparable to the baseline. Over the entire course of training, LENS results in only a 6.5% increase in total wall-clock time.
>
> > Can the authors provide a theoretical justification for the number 2 in the
> $D(q)$ estimator 1 (line 370) and explain why the estimator must be defined differently for mixed vs. negative groups?
>
> Please refer to General Response 2 for the detailed rationale. The split definition is strictly necessary because the Importance Sampling estimator is undefined in negative groups (where no correct answer exists to anchor the difficulty estimate). regarding the factor 2: it serves as a hyperparameter for numerical stability.
>
> We thank the reviewer once again for the time and effort. Please let us know if we have addressed all the questions.

---

### Official Review · Reviewer_npA8 · 2025-11-09

**Soundness:** 3
**Presentation:** 3
**Contribution:** 2
**Rating:** 6
**Confidence:** 3

**Summary:**

This paper focuses on solving the issue of original GRPO objective that the negative groups contribute no gradient to the update. The authors propose a method called Likelihood Estimation with Negative Samples (LENS), which assigns non-zero advantage to responses within negative groups based on a confidence-weighted penalty. Experiments on two models verify the effectiveness of the proposed GRPO-variant objective.

**Strengths:**

- The paper provides a conceptually clear connection between reward modeling and policy learning.
- The derivations are rigorous, with clear assumptions and logical consistency.

**Weaknesses:**

- Can the authors prove that results in Table 1 are statistically significant (at least 2-sigma or 95% confidence level deviations)?
- Only one benchmark is evaluated in this paper, how does the methods generalize to more benchmarks (e.g., AIME, AMC, etc.)?

**Questions:**

1. The proposed method modifies the GRPO objective by introducing a confidence penalty to the negative groups, I'm wondering how this connects with the entropy/confidence-related methods like [1][2][3].
2. [4] shows that simply assigning -1 rewards to negative responses and dropping the correct ones can improve LLM reasoning, what if the proposed objective in this paper is applied to negative-only groups and dropping other samples? Will this be better than simply assigning -1 rewards?

[1] Learning to Reason without External Rewards

[2] Beyond the 80/20 Rule: High-Entropy Minority Tokens Drive Effective Reinforcement Learning for LLM Reasoning

[3] Maximizing Confidence Alone Improves Reasoning

[4] The Surprising Effectiveness of Negative Reinforcement in LLM Reasoning

---

> ### Author Response · Authors · 2025-11-25
>
> We sincerely thank the reviewer for their feedback and for recognizing the clarity of our theoretical logic. We address the specific concerns as follows.
>
> > how this connects with the entropy/confidence-related methods like [1][2][3].
>
> Thanks for this question. While we share the intuition that model confidence is critical, our work differs fundamentally in derivation and granularity:
>
> Compared with [1, 3]): Our objective is not an ad-hoc regularization term. It is mathematically derived by connecting reward modeling with policy optimization; the confidence-based penalty emerges naturally from this dual view. In contrast, [1] relies on a heuristic constraint (KL divergence to a uniform distribution, which is pure heuristic), and [3] designs a sequence-level reward using average entropy. Our formulation provides a theoretical justification for why confidence minimization is optimal in this setting.
>
> Compared with [2]: [2] utilizes token-level entropy specifically for masking within GRPO to save unnecessary optimization. Our method operates on sequence-level confidence directly within the optimization objective. The motivation is different.
>
> > Error bars and more evaluation dataset.
>
> Please see General Response 1.
>
> > [4] shows that simply assigning -1 rewards to negative responses and dropping the correct ones can improve LLM reasoning, what if the proposed objective in this paper is applied to negative-only groups and dropping other samples? Will this be better than simply assigning -1 rewards?
>
> Please see General Response 3.
>
> We thank the reviewer once again for the time and effort. Please let us know if we have addressed all the questions.

---

### Author Response · Authors · 2025-11-25
**General Response**

We sincerely thank all reviewers for their careful consideration and the time invested in reviewing our work. We are encouraged that the reviewers recognized the value of our method in addressing real-world data inefficiencies in GRPO, as well as the novelty of our theoretical insights bridging reward modeling and policy gradients.

We aim to cover some general concerns here. Concern 1 is shared across Reviewer npA8, bfBj, and y75z; Concern 2 is shared across Reviewer 285B and bfBj; Concern 3 is shared acorss npA8, 285B, bfBj, and y75z. We also updated the manuscript and marked the changes in blue.

1. > Error bars and more evaluation dataset.

Following your suggestion, we have now performed an extensive additional suite of experiments. We perform 3 runs and estimate the std for Table 1 while extending the evaluation to include not only MATH, MATH level 4/5, but also OlympiadBench, MinervaMath, GSM8k. The results are updated in the main paper in blue. The copy the results here:

**Table 1:** Comparison of our method against the baseline using `Qwen-2.5-3B-Base`. Values denote accuracy (%) Mean ± Std. Generated with 2 random seeds.

| Evaluation Set | GRPO Baseline | LENS (Ours) |
| :--- | :---: | :---: |
| MATH | 66.11 ± 0.38 | **68.35 ± 0.67** |
| MATH Levels 4-5 | 49.09 ± 0.26 | **51.82 ± 0.35** |
| GSM8K | 85.61 ± 0.12 | **85.98 ± 0.16** |
| MinervaMath | 26.67 ± 0.45 | **27.44 ± 0.26** |
| OlympiadBench | 30.91 ± 0.24 | **32.78 ± 0.27** |

This confirms that: (1) performance is stable across seeds with negligible deviation when we scale RL to thousands of steps; and (2) LENS achieves statistically significant improvements over GRPO on MATH, MATH Levels 4–5, MinervaMath, and OlympiadBench.

2. > If the theoretical framework is a loose inspiration rather than a principled derivation.
> Breakdown of the MLE/IS Derivation for Negative Groups. This weakens the claimed theoretical connection between reward modeling and policy optimization, especially in emphasizing resolving the learning signal from the incorrect group.

We thank the reviewer for this thoughtful comment. We agree that for all-negative groups, the direct Importance Sampling estimator is undefined. However, we respectfully argue that the fallback is not merely "ad-hoc," but follows a standard machine learning practice of treating theoretical constraints as tunable hyperparameters for empirical stability—analogous to the penalty coefficient $\lambda$ in Ridge Regression.

In Ridge Regression, the penalty coefficient $\lambda$ has a theoretical derivation (e.g., related to the Lagrange multiplier of a norm constraint or the variance of a Bayesian prior). However, in practice, $\lambda$ is almost always treated as a tuning parameter to balance bias and variance.

Similarly, in our framework: theoretically, $D(q)$ is formally defined as the normalization factor $\{\sum p^* \}^{-1}$. In practice, functionally, $D(q)$ controls the sensitivity of the negative reinforcement. As shown in Eq. (10) 2, the calibrated reward scales with $\frac{\pi}{D - \pi}$. If $D$ is set very large (conservative), the penalty vanishes (under-fitting the negative signal). If $D$ is set very close to $\pi$ (aggressive), the penalty explodes (instability).

Therefore, in the absence of empirical positive samples (negative groups) to estimate the "true" $D$, we treat $D(q)$ as a dynamic hyperparameter governed by the scaling factor $\beta$:$$D(q) = \beta \cdot \max_i \pi(o_i|q).$$ We empirically chose $\beta=2$ as a robust setting that acts as a "safety margin." It ensures the penalty is strong enough to be informative but bounded enough to maintain numerical stability.

The claimed theoretical connection holds: the form of the objective (using a confidence-based penalty $\frac{\pi}{D-\pi}$) is derived from MLE. The value of $D(q)$ in negative groups is approximated via a hyperparameter to operationalize this theory in the wild, much like selecting a regularization strength.

---

> ### Author Response · Authors · 2025-11-25
> **General Response 2**
>
> 3. > Comparison with the NSR baseline (Zhu et al. 2025).
>
> We omitted the NSR baseline (Zhu et al., 2025) primarily due to severe training instability. In our experiments across different backbones (e.g., Qwen, Llama), NSR maintains performance for fewer than 150 optimization steps. Beyond this point, the models suffer from collapse, characterized by monotonically increasing response lengths and a sharp degradation (to 0%) in generation quality. This is also noted in the original paper in the appendix Section I, "We observe that extensive training with NSR (e.g., over hundreds of gradient steps) leads to a noticeable decline in performance".
>
> While specific hyperparameters might yield some improvements in this short range, we argue such results offer limited practical significance for guiding the scaling of RL. As advocated by recent work (Khatri et al., 2025), the utility of RLVR is better measured by stability and scaling potential rather than peak performance in early training. Therefore, we prioritized comparisons with stable algorithms that support sustained optimization.
>
> Khatri D, Madaan L, Tiwari R, et al. The art of scaling reinforcement learning compute for llms[J]. arXiv preprint arXiv:2510.13786, 2025.

---

### Meta-Review · Area_Chair_pT79 · 2026-01-07

**Summary:**

As a first, significant note: you should _not_ have included the acknowledgements section in your revision! As this breaks anonymity, it does serve as grounds for automatic rejection. Especially given the circumstances of this highly unusual ICLR in which anonymity was perhaps broken anyway, I am not taking any action based on this lapse, but ensure you do not make the same mistake in the future.

----

I agree with the reviewers that it was concerning the initial submission considered only a single dataset. The rebuttal experiments considering several other math datasets are appreciated, although since nothing in the method is actually math-specific, it would have been nicer to also consider other RLVR settings like code generation or logical reasoning. (The "knights and knaves" dataset is a simple one of a somewhat different form from mathematical problems that is common in this area.) I appreciate that the initial submission also considered Llama models; having both Llama and Qwen significantly improves the confidence that the method is reasonably general and not "Qwen-specific" for some reason.

----

Another note that is I think somewhat significant: in your response to reviewer bfBj, you said

> In this formulation, the normalization term becomes $D(q)=(\int_{\mathcal O} p^\star(q,o),\rho(q,do))^{-1}$, which is well defined because $p^\star(q,o)\in[0,1]$ is uniformly bounded.

This is not true. Consider the response-length preference function $\rho(q, o) = \gamma^{| o |}$ you discuss in Appendix C, and suppose that $\mathcal O$ is (as it actually essentially is in practice) the set of arbitrary-length strings over an alphabet $\mathcal A$. Also suppose, for simplicity, that all answers are correct: $p^*(q, o) = 1$. Then $1/D(q) = \sum_{o \in \mathcal O} \gamma^{|o|} = \sum_{n=0}^\infty (| \mathcal A | \gamma)^{n}$. This geometric series converges iff $| \mathcal A | \gamma < 1$, i.e. iff $\gamma < 1 / | \mathcal A |$; for larger $\gamma$, $D(q)$ is undefined. This does not matter in practice since you don't directly use $D$ but rather a rough guess based on a "scaling factor," but is still somewhat concerning.


----

I somewhat share reviewer 285B's concern about

> doesn't this imply the theoretical framework is a loose inspiration rather than a principled derivation?

Compared to the case of $\lambda$ in ridge regression: you say it has a

> theoretical derivation (e.g., related to the Lagrange multiplier of a norm constraint or the variance of a Bayesian prior). However, in practice, is almost always treated as a tuning parameter to balance bias and variance

Neither the norm constraint nor the variance of a prior are a "derivation" for the value of $\lambda$! The norm constraint is a different way to balance bias and variance (and cannot be converted to a $\lambda$ a priori), while choosing the variance of a prior is exactly equivalent to choosing $\lambda$ and itself a hyperparameter that must be set somehow.

By contrast, $D(q)$ has a very clear definition in your derivation that you estimate from data through importance sampling when possible. For all-negative groups, you come up with a heuristic to guess at it. This is indeed standard practice – you have to do _something_ – but it's very believable that the choice you make here has a strong influence on the behavior of the method in the presence of all-negative groups, and you don't substantially explore that. In particular, I would strongly encourage you to do some variation of the following experiment: take a prompt with a low probability of reward, and sample many times. Then consider hypothetical samplings for some reasonable size which are all negative, versus those that contain some positive examples. How consistent is the estimate $D(q)$ across the all-negative cases versus the cases including some positives?

For that matter, how good is the $D(q)$ estimate even in cases where you get a positive? Importance sampling can have notoriously high variance; is that affecting your final estimation quality? It's unsatisfying to have these important sub-estimators in your method, but no exploration of how your approaches there correspond to the mathematical motivation, just an idealized model and then experiments on RLVR quality (which involve many more factors than just $D(q)$).

---

Overall, I like the idea proposed here, and think that the mathematical motivation is satisfying. I do still agree with reviewer bfBj that "the paper is good on the theoretical side and could be further improved on the experimental side." While the rebuttal certainly took steps in that direction, I strongly encourage the authors to go further in their exploration of the ideas underlying their approach and experimentation before resubmitting to a future venue.

**Reviewer Concerns:**

Discussed above.

**Reviewer Scores:**

I think there likely would have been some mild increases, but nothing dramatic, based on the concerns and the responses.

---

### Decision · Program_Chairs · 2026-01-26

Reject